# 2D/2D coupled MOF/Fe composite metamaterials enable robust ultra–broadband microwave absorption

Ning Qu[1,2], Hanxu Sun[1,2], Yuyao Sun[1], Mukun He[1], Ruizhe Xing ®[1] ✉, Junwei Gu[1] & Jie Kong ®[1] ✉

The combination between macroscopic structure designs and microscopic material designs offers tremendous possibilities for the development of advanced electromagnetic wave (EMW) absorbers. Herein, we propose a metamaterial design to address persistent challenges in this field, including narrow bandwidth, low–frequency bottlenecks, and, particularly, the urgent issue of robustness (i.e., oblique, and polarized incidence). Our absorber features a semiconductive metal·organic framework/iron 2D/2D assembly (CuHT–FCIP) with abundant crystal/crystal heterojunctions and strong magneto-electric coupling networks. This design achieves remarkable EMW absorption across a broad range (2 to 40 GHz) at a thickness of just 9.3 mm. Notably, it maintains stable performance against oblique incidence (within 75°) and polarizations (both transverse electric and transverse magnetic). Furthermore, the absorber demonstrates high specific compressive strength (201.01 MPa·cm$^3$·g$^{-1}$) and low density (0.89 g·cm$^{-3}$). This advancement holds promise for developing robust EMW absorbers with superior performance.

The widespread adoption of fifth-generation (5 G) technology implies a magnitude increase in the power of electromagnetic waves (EMWs) in the space, significantly elevating the need to balance essential monitoring activities with privacy concerns and remediating health issues arising from prolonged exposure[1]. A suitable EMW absorber plays an important role in resolving such issues. Conventional EMW absorption materials achieve absorption at the resonant thickness by manipulating the dispersion properties[2]. Nevertheless, their effective absorption bandwidth (EAB) is constrained to a narrow range due to the limitations of causality[3]. Metamaterial absorbers, by artificially constructing structures at subwavelength scales, demonstrate infinite potential in extending the EAB of existing materials, particularly in the terahertz[4], infrared[5], and visible light spectra[6]. Nevertheless, achieving perfect absorption over an ultra–broad microwave frequency range, covering the entire 5 G communication band, still remains most challenging due to the long wavelengths and high penetrability of the

microwaves[7]. The recently emerged three–dimensional (3D) metamaterials, leveraging the 3D topological structure, have showcased promising bandwidth by further incorporating dielectric/magnetic losses and diffraction resonances[8]. However, the performance of 3D metamaterials relies heavily on the EMW loss properties of the materials themselves[9], with the most optimal performance typically include complex micro/nanostructures (e.g., multi-core-shell nanoparticles[10], nanoflowers[11] and hierarchical nanoparticles[12]). This greatly compromise the stability of magnetoelectric coupling of materials and, hence, the robustness (i.e., oblique, and polarized incidence) of absorption for both experimental implementation and mass production. Therefore, it is crucial to explore materials with simple/stable morphology but strong EMW loss to facilitate the commercialization of 3D metamaterials.

Metal–organic frameworks (MOFs), constructed by coordinating inorganic metal nodes with organic ligands, have several appealing

[1]Shaanxi Key Laboratory of Macromolecular Science and Technology and MOE Key Laboratory of Materials Physics and Chemistry in Extraordinary Conditions, School of Chemistry and Chemical Engineering, Northwestern Polytechnical University, Xi'an 710072, P. R. China. [2]These authors contributed equally: Ning Qu, Hanxu Sun. ✉e-mail: rzxing@nwpu.edu.cn; kongjie@nwpu.edu.cn

features including high surface area, tunable active sites, and controllable crystal structure (e.g., rods, sheets, or polyhedron)[13]. The majority of metals can be selected to obtain MOFs (especially magnetic metals, such as iron, cobalt, and nickel), which makes them promising candidates in EMW absorption[14]. Plus, the porous nature of MOFs provides abundant active sites for interactions with the surrounding molecules or ions, effectively enhancing the Maxwell-Wagner polarization, thus leading to enhanced EMW loss ability[15]. Unfortunately, MOFs typically exhibit poor electrical conductivity because the linking organic groups are mostly insulators with small π-orbital conjugation[16]. Therefore, most conventional MOFs show EMW absorption capabilities only after high-temperature pyrolysis, during which the ligand framework is transformed into metal-doped carbon materials[17]. This not only leads to the densification of MOF materials but also introduces extra uncertainty in the topology of MOF-derived structures[18].

Semiconductive MOFs (SC-MOFs) show significantly improved intrinsic conductivity by enhancing the conjugation on the organic linker and utilizing electron-rich metal/organic linkers[19]. The low bandgap and tunable active sites of SC-MOFs enable efficient interface charge transfer mobilities, injecting significant momentum to catalysis[20] and energy fields[21], while also demonstrating tremendous potential in EMW absorption[22]. However, most SC-MOFs still show moderate magnetism because the spin coupling of metal ions is constrained within the MOF structure, and the presence of large-size ligands promotes the establishment of random spin arrangements[23]. As a result, SC-MOFs are highly susceptible to an imbalance in magnetoelectric coupling, leading to poor impedance matching performance[24]. From these perspectives, the incorporation of SC-MOFs with strong magnetic materials could allow for the rational design of a superior magnetoelectric coupling effect, which not only enhances the EMW absorption capabilities of SC-MOFs but also expands the EAB further when applied in 3D metamaterials. In

addition, the stable and tunable crystal morphology of SC-MOFs, especially their two-dimensional (2D) sheet-like structure, attendant with a pyrolysis-free preparation method, opens up new opportunities for the development of efficient and robust EMW metamaterial absorbers.

In this work, we successfully fabricated a 2D micro-sheet of SC-MOF (CuHT) and integrated it with flake-layered carbonyl iron powders (FCIPs), resulting in the formation of a sandwich-like 2D/2D assembly (CuHT-FCIP) (Fig. 1). 2D materials have thin thickness below skin depth, which could break the Snoek limit and enhance the initial magnetic permeability[25]. Moreover, theoretical simulation revealed that the crystal/crystal heterojunction of CuHT-FCIP facilitates the extraction of electrons from the CuHT surface by the C atoms belonging to FCIP, which promotes the generation of numerous interlayer charge-transfer pathways and thus leads to a strong magnetoelectric coupling loss. Furthermore, we encapsulated CuHT-FCIP in an electromagnetic transparent epoxy resin (CuHT-FCIP-EP) and achieved an effective modulation of the EAB in the range of 2 GHz to 18 GHz. Finally, we integrate CuHT-FCIP-EP into a novel 3D metamaterial design, which combines a gradient impedance design with honeycomb perforation. The as-prepared CuHT-FCIP-EP metamaterials show a robust microwave absorption performance, with an ultra-broad EAB from 2 GHz to 40 GHz, ultra-wide oblique adaptability within 75°, and high polarization insensitivity. In addition, the straightforward manufacturing process and excellent mechanical performance make it well-suited for mass production and development of functional ultra-broadband EMW absorbers.

## Results
### Preparation of CuHT-FCIP composites
To prepared CuHT-FCIP composites, we first fabricated 2D semiconductive MOF (CuHT) and flake-layered carbonyl iron powders (FCIPs). CuHT was synthesized using a modified route to precisely

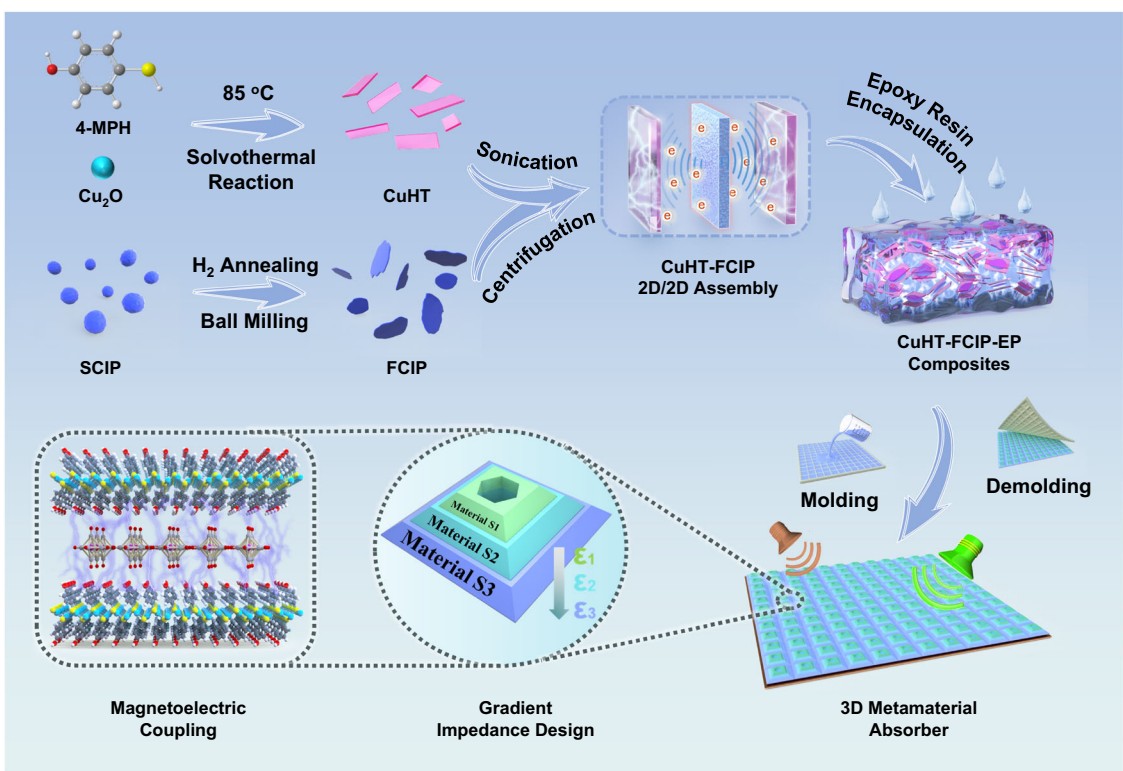

**Fig. 1 | Schematic illustration of the preparation of CuHT-FCIP-EP metamaterial absorber.** The CuHT-FCIP 2D/2D assembly is fabricated by immersing the CuHT and FCIP mixture in an ethanol solution and subjecting it to bath sonication.

Next, the CuHT-FCIP assembly is encapsulated in epoxy resin at a specific ratio. Finally, different proportions of CuHT-FCIP epoxy resin are sequentially cured in the designed metamaterial mold to produce CuHT-FCIP-EP metamaterials.

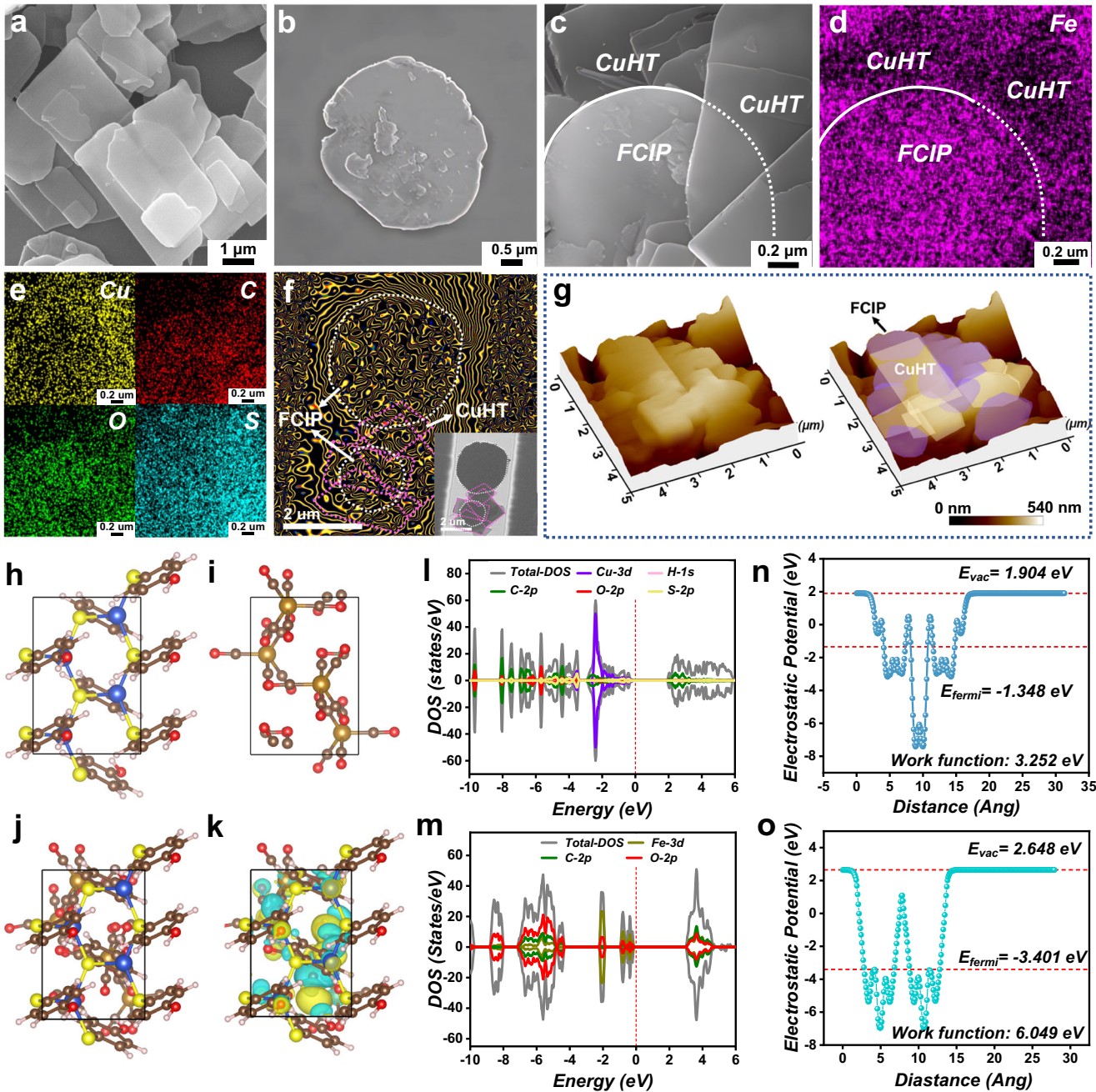

**Fig. 2 | Structural characterization of CuHT–FCIP composite.** SEM images of **a** CuHT, **b** FCIP, **c** CuHT–FCIP composites. **d–e** Corresponding element mapping results of Fe, Cu, C, O and S. **f** Magnetic flux line distribution of the FCIP–CuHT sandwich composite calculated from the corresponding off-axis hologram (Inset). **g** AFM images of CuHT–FCIP sandwich composite. The view of **h** CuHT, **i** FCIP and **j** CuHT–FCIP composites structures along the c-axis. **k** Differential charge density of CuHT–FCIP between contact surfaces (Turquoise and bright yellow colors represent the decrease and enrichment of electron, respectively). The density of states of **l** CuHT and **m** FCIP. The work function of **n** CuHT and **o** FCIP.

control the growth and nucleation along the specific crystal plane. The resulting CuHT products have a rectangular 2D flake morphology with an average length to width ratio of around 1.70 (Fig. 2a and Supplementary Fig. 1). In addition, CuHT exhibits typical semiconductor properties, with a bandgap of 2.87 eV and a bulk conductivity of $4.23 \times 10^{-3}$ S·cm$^{-1}$ (Supplementary Fig. 2). Commercial spherical carbonyl iron powders (SCIPs) were ball milled to obtain FCIPs. The resulting FCIPs then underwent hydrogen annealing treatment prior to utilization. This treatment was employed to enhance magnetism by reducing the presence of admixtures such as $Fe_3N$, $Fe_3C$, and $Fe_3O_4$. The finally obtained FCIPs have a circular wafer shape with a diameter of about 4.05 μm, which closely matches the dimension of CuHT

(Fig. 2b). This unique morphology matching characteristic also indicates an excellent compatibility between CuHT and FCIP in terms of their magnetoelectric performance.

The packing density of CuHT is only 0.0856 g·cm$^{-3}$, which is merely 1.09% of that of FCIP. Therefore, simply mixing them would inevitably result in stratification, thereby failing to establish coupling between CuHT and FCIP. In this work, we tackle this issue by introducing ultrasonic energy into the system. During sonication in ethanol, Cu ions from the surface of CuHT begin to dissolve, leaving uncoordinated thiol (−SH) terminals. As evidenced by the X-ray photoelectron spectroscopy (XPS) results, these −SH groups then bond with Fe in the FCIP, forming much more stable Fe−S bonds (Supplementary

Fig. 3). The resulting CuHT–FCIP composites are uniformly dispersed in the ethanol solution (Supplementary Fig. 4). Scanning electron microscopy (SEM) image shows that CuHT–FCIP composite exhibit a typical layered microstructure (Fig. 2c). The atomic force microscopy (AFM) image further confirms the sandwich-like structure where CuHT and FCIP are layered with interspersed regions of differing composition. (Fig. 2g). The energy dispersive X-ray spectroscopy (EDS) mapping results reveal the uniform distribution of the various elements inside the composite material (Fig. 2d, e). In addition, electron holography results show that stray magnetic flux lines not only radiate into the free space to form long-range magnetic interaction but also interact with neighboring FCIP domains to achieve multi-domain coupling via flux linkage arrangements (Fig. 2f), realizing a magnetically connected coupling network. This lays the foundation of magnetic loss in the CuHT–FCIP composite.

The 2D/2D coupling network in CuHT–FCIP composite effectively increases the contact area of the sheet materials, facilitating the formation of electron transport pathways. To prove this, we performed density-functional theory (DFT) calculations[26] to analyze the charge-transfer mechanism at the crystal/crystal heterogeneous interface between CuHT and FCIP. The CuHT and FCIP modes were first constructed according to the previous structural information (Fig. 2h, i). The density of states (DOS) is used to reveal the electron transfer in the CuHT–FCIP composite. The DOS intensity of CuHT is higher than that of FCIP, indicating the accumulation of electrons in CuHT (Fig. 2l, m). A great number of electrons tend to migrate from CuHT to FCIP rather than the opposite. In addition, when two materials are combined, it is generally believed that electrons migrate more easily from the material with the lower work of function ($\Phi$) to the material with the higher $\Phi$[27]. Based on this, the calculated $\Phi$ for CuHT and FCIP are 3.252 eV and 6.049 eV, respectively (Fig. 2n, o). Therefore, the electrons on CuHT will migrate towards FCIP to reach the balance of Fermi level.

The differential charge density[28] can intuitively and clearly reveal the charge distribution on the CuHT–FCIP composites. Compared with the status before combination (Fig. 2j), the charge density rearranged after the integration of CuHT with FCIP (Fig. 2k). It can be clearly seen that the electrons are extracted from the surface of CuHT by C atoms belonging to FCIP. This phenomenon once again proves the electron affinity of FCIP, which facilitates charge transfer between heterogeneous interfaces and gives rise to an interfacial polarization, thereby promoting the establishment of a magnetoelectric coupling network and enhancing the EM loss ability of the material.

## Electromagnetic performance of CuHT–FCIP–EP composites

Benefiting from the abundant interlayer magnetic and electric coupling effects of CuHT–FCIP composites, we further encapsulate them with an electromagnetic transparent epoxy (EP) resin to evaluate their EMW absorption performance. The reflection losses (RLs) for composite materials under different mass ratios (Supplementary Table 1) are presented in Fig. 3 and Supplementary Fig. 5. Sample containing only CuHT (CuHT–EP) demonstrates an effective absorption bandwidth (EAB) of only 2.61 GHz. This is mainly attributed to the electrical conduction loss of CuHT itself and the abundant interfacial polarization between the 2D layered structures (Supplementary Fig. 6)[29]. Despite the moderate performance, this indicates the potential of semiconductive MOFs in EMW absorption capabilities. FCIP–EP exhibits poor impedance matching at a low mass fraction of 30% (EAB = 1.03 GHz). However, as the content of FCIP increases to 70%, there is a notable improvement in EMW absorption performance, with the EAB reaching 5.02 GHz. Particularly noteworthy is the achievement of a minimum reflection loss (RL$_{min}$) of −55 dB (over 99.999% dissipation), especially in the low-frequency range (2–6 GHz). This is primarily attributed to the excellent ferromagnetic properties[30] as well as the quasi-two-dimensional layered morphology of FCIP, which greatly

enhances the interlayer magnetic natural/exchange resonance loss (Supplementary Fig. 7). On the other hand, the low electrical conductivity of FCIP itself results in a mismatch in magnetoelectric coupling either when added in excess or insufficient amounts (Supplementary Fig. 8). Therefore, the FCIP content could be crucial in modulating the overall absorption performance of the CuHT–FCIP composite system.

To further explore the effect of FCIP content on the EMW absorption performance of the CuHT–FCIP–EP composite system, we prepared three samples with different CuHT/FCIP ratios: 4:2 (CuHT–FCIP–EP–S1), 3:3 (CuHT–FCIP–EP–S2), and 3:4 (CuHT–FCIP–EP–S3). As depicted in Fig. 3a–c and Supplementary Fig. 9, the CuHT–FCIP–EP composite system demonstrates significantly enhanced EMW absorption performance compared to individual components. Specifically, CuHT–FCIP–EP–S1 exhibits an EAB of 5.44 GHz and a RL$_{min}$ value of −52 dB (over 99.999% dissipation) at 14.6 GHz. With an increased FCIP/CuHT ratio, the overall strongest absorption position shifts towards lower frequencies, and the corresponding thickness of the maximum EAB gradually decreases. This is mainly attributed to the ferromagnetic properties of FCIP, which significantly enhance the magnetic loss for EMW according to classic transmission line theory[31]. CuHT–FCIP–EP–S2 exhibits an EAB of 6.16 GHz and an RL$_{min}$ of −61 dB (over 99.9999% dissipation) at 5.2 GHz, indicating that the 2D/2D magnetoelectric coupling effect of CuHT–FCIP peaks at this ratio.

Dielectric loss factor (tan $\delta_\varepsilon$) and magnetic loss factor (tan $\delta_\mu$) quantify the efficiency of electric and magnetic energy dissipation when subjected to an alternating electromagnetic field[32]. The overlapping peaks in the frequency-dependent dielectric/magnetic loss factor diagram indicate a strong magnetoelectric coupling loss, which typically signifies a strong EMW absorption at the corresponding frequency. As shown in Fig. 3d–f, both the dielectric and magnetic loss factor curves exhibit three prominent peaks, approximately at 2–8 GHz, 8–14 GHz, and 14–18 GHz. For CuHT–FCIP–EP–S1, the peak positions in the 2–8 GHz range are nearly aligned. In CuHT–FCIP–EP–S2, opposite trends are observed in the 2–6 GHz and 16–18 GHz ranges. Similarly, for CuHT–FCIP–EP–S3, the peak trends in the 12–18 GHz range are almost opposite. These results indicate that adjusting the ratio of CuHT/FCIP allows for effective control of the magnetoelectric matching frequencies, which enables the EMW energy dissipation at a given frequency range.

The calculated surface impedance[33] results reveal that S1, S2, and S3 have an impedance equal to that of air (i.e., $|Z_{in}/Z_0| = 1$) at 2.9 GHz, 5.0 GHz, and 14.0 GHz, respectively (Fig. 3g). This is consistent with the magnetoelectric coupling analysis, suggesting that the 2D/2D coupling of CuHT–FCIP can effectively achieve both strong electromagnetic energy loss and surface impedance matching. The radar cross section (RCS) simulation[34] results for a 200 mm × 200 mm × 2.25 mm flat panel indicate that CuHT–FCIP–EP–S2 can reach a maximum RCS reduction of up to 25 dBm$^2$ compared to a perfect electric conductor (PEC). This highlights the excellent electromagnetic wave (EMW) stealth performance of CuHT–FCIP–EP (Fig. 3h). Compared to other recently reported MOF-derived carbon materials, CuHT–FCIP–EP demonstrates unparalleled advantages in terms of EAB, thickness, and maximum absorption intensity (Fig. 3i). More importantly, it offers a pyrolysis-free preparation process, distinguishing it further.

## Design of the CuHT–FCIP–EP metamaterial

Although the 2D/2D magnetoelectric coupling of CuHT–FCIP sandwich-like structure endows the material with an excellent EMW attenuation performance. The EAB of CuHT–FCIP–EP composite remains suboptimal due to the constraint of causality[35]. In addition, dispersion-based EMW absorbing materials undergo changes in refraction energy and optical path when the incident angle varies widely, which leads to a significant decrease in absorption efficiency[36].

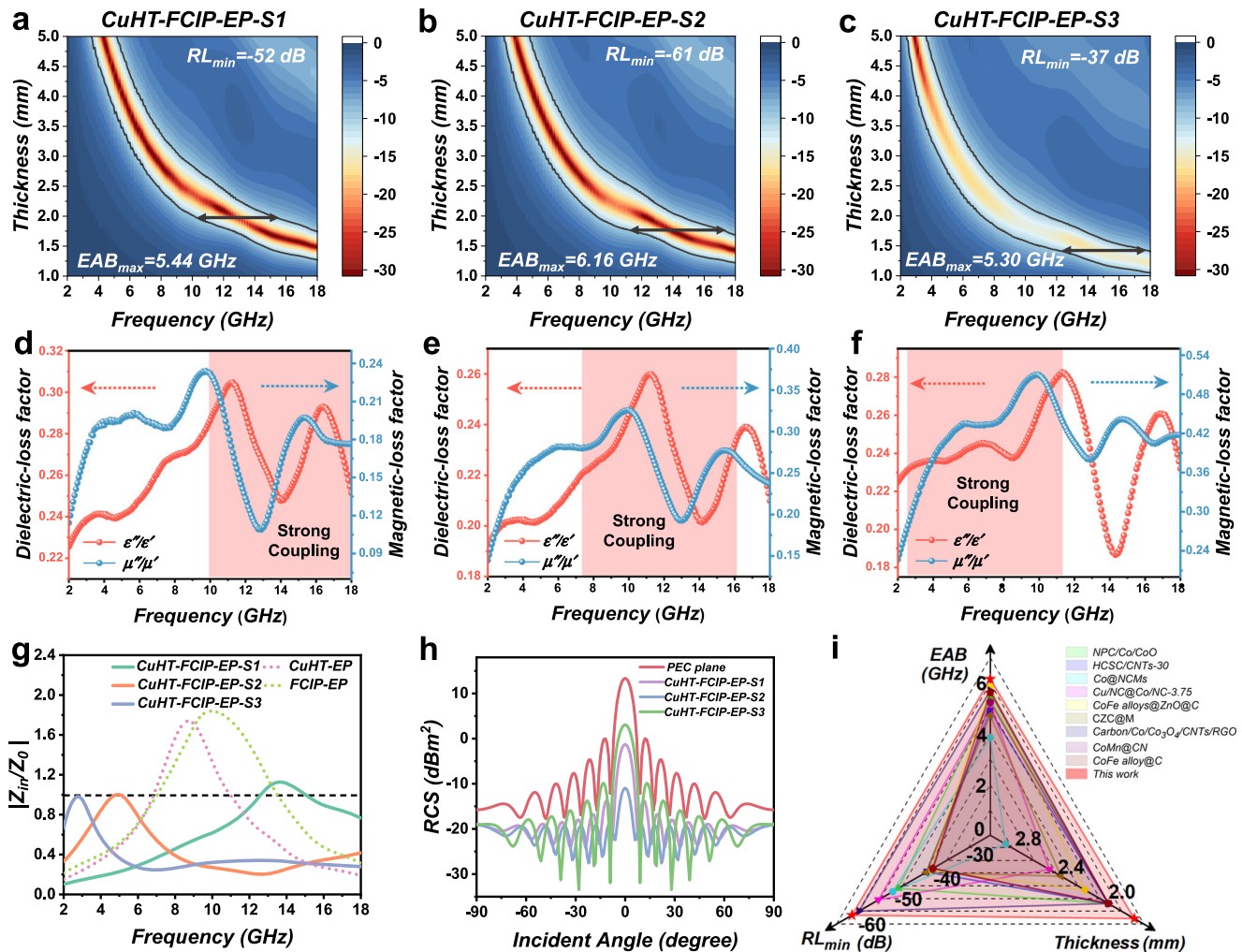

**Fig. 3 | EMW absorption performance of CuHT−FCIP−EP composite. a−c** 2D contour diagrams of RL values versus frequency at different thicknesses for samples S1−S3. Source data are provided as a Source Data file. **d−f** Dielectric loss (tan $\delta_\varepsilon$, red) and magnetic loss (tan $\delta_\mu$, blue) curves of sample S1−S3. Source data are provided as a Source Data file. **g** Relative surface impedance ($|Z_{in}/Z_0|$) of sample S1−S3. Source data are provided as a Source Data file. **h** Simulated RCS values of samples under various incident angles at 10 GHz. **i** Comparison of EMW absorption properties with recently reported composites derived from MOFs. The relevant data are summarized in Supplementary Table 2.

Recently reported 3D metamaterials integrate the advantages of traditional 2D metamaterials and absorbing materials[37]. These materials not only directly attenuate EMW energy but also achieve further attenuation through structural resonances and secondary reflections from the cavity structure. As a result, they can surpass the causality−dictated minimum thickness and significantly enhance the EAB of the original materials[38]. In addition, due to the introduction of topological structures, the absorption deterioration under oblique and polarized incidence conditions is also much smaller than that of traditional flat absorption materials. Therefore, it is of great potential to combine the as-prepared CuHT−FCIP with proper 3D metamaterial structure design.

We propose a metamaterial design approach that combines gradient impedance matching layer with honeycomb perforation, successfully broadening the EAB for CuHT−FCIP−EP composite. The unit cell consists of three layers of materials (S1−S3) stacked together. They collectively form a truncated pyramid structure with honeycomb perforations at the top (Fig. 4a). The overall unit cell is defined by 12 parameters: seven horizontal parameters (L1−L7) and five vertical parameters (T1−T5). We first optimized the arrangement of the three layers by creating a periodic 10 mm × 10 mm unit structure. The results reveal that when arranged from top to bottom as S1−S2−S3, the power

loss contributions of the overall material perfectly match and connect across the low-frequency range (2−6 GHz), mid-frequency range (6−18 GHz), and high-frequency range (18−40 GHz). (Fig. 4b) This corresponds to the characteristic impedance matching frequency of each layer material in Fig. 3g and eventually ensures a broadband absorption across the entire 2−40 GHz range (Supplementary Fig. 10, Supplementary Table 3). Next, various structural parameters of the metamaterial are optimized by simulation using the EAB as the objective function (Supplementary Fig. 11). Ultimately, we finalized the structural parameters of the metamaterial (Supplementary Table 4) and fabricated the CuHT-FCIP-EP metamaterial test sample measuring 180 mm × 180 mm (Fig. 4a inset).

To better understand the attenuation mechanism of the CuHT−FCIP−EP metamaterials, the surface energy flow densities and electric/magnetic field intensities of the unit structure are analyzed (Fig. 4c and Supplementary Fig. 12). According to the simulated reflection curves, the CuHT−FCIP−EP metamaterial absorber show four prominent absorption peaks within the frequency range of 2−40 GHz. Specifically, the peaks at 16.1 GHz and 32.2 GHz are mainly attributed to thickness resonances at λ/2 and λ (wavelength), while the peaks at 3.82 GHz and 22.0 GHz are primarily attributed to the resonance of the bottom layer (S3) and the honeycomb structure[39]. (Fig. 5a

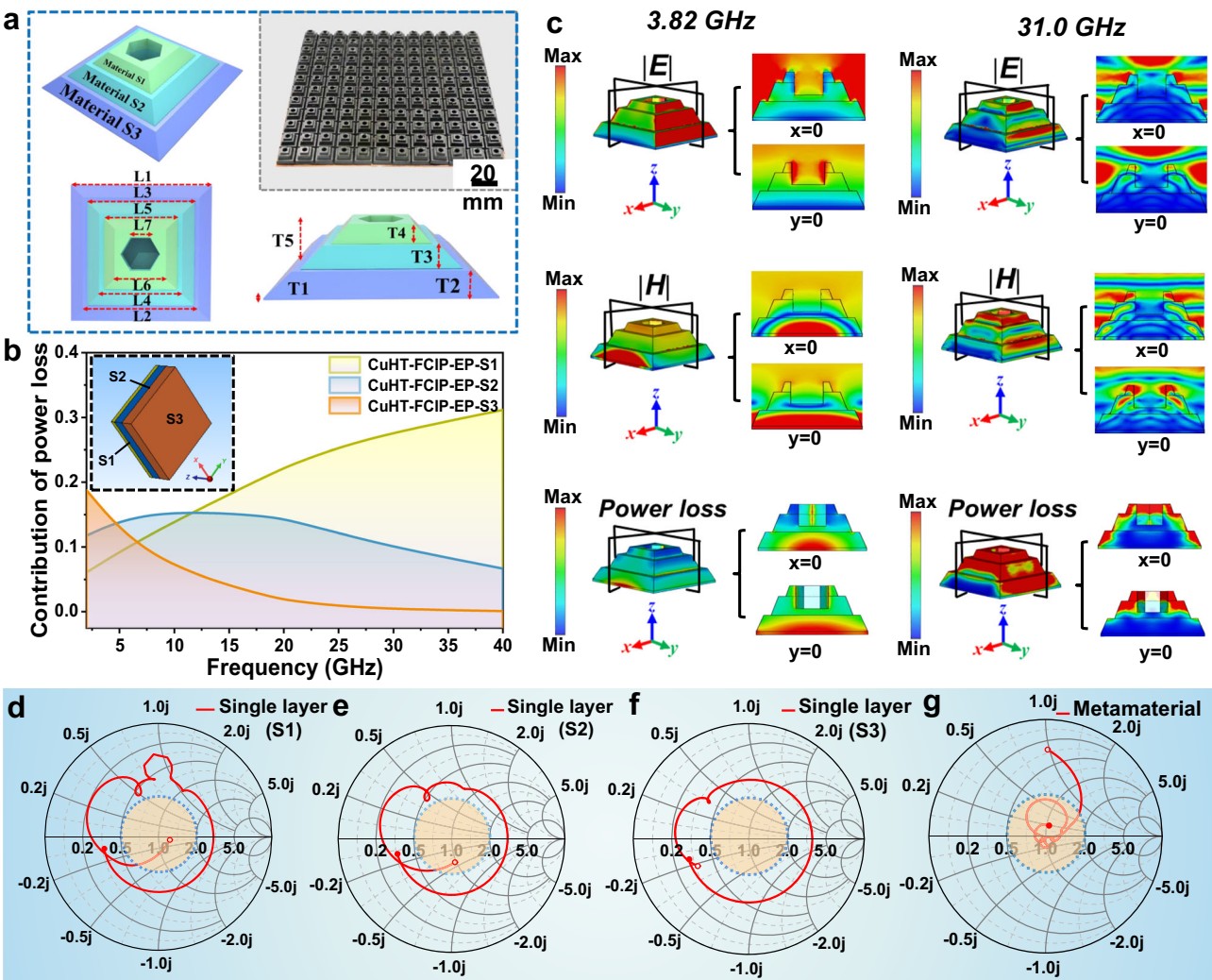

**Fig. 4 | Design of ultra-broadband metamaterial absorber. a** Schematic diagram of the proposed absorber, structural parameters, and the as-prepared metamaterial sample. **b** Frequency-dependent power loss contribution for the three-layered slab model (S1–S2–S3) derived from electromagnetic finite elements simulation. Inset: the simulation model (10 × 10 mm unit cell). Source data are provided as a Source Data file. **c** Distributions of the electric field, magnetic field, and power loss density of the metamaterial absorber at 3.82 GHz and 31.0 GHz. **d–g** Smith chart of single-layer absorbers and metamaterial absorbers (the hollow circle end represents 2 GHz, while the solid end represents 40 GHz; the orange shadowed region represents a fluctuation range of 31.6% around the normalized impedance near perfect matching impedance (i.e., |Z| = 1). Source data are provided as a Source Data file.

and Supplementary Fig. 13). The honeycomb perforation enables increased entry of low-frequency EMWs. Consequently, at 3.82 GHz, the electric and magnetic fields exhibit a concentrated distribution in the bottom layer of the metamaterial. This suggests that the bottom layer material (i.e., CuHT-FCIP-EP-S3) primarily functions in absorbing EMWs at the low-frequency range (Supplementary Fig. 14). As the wavelength decreases, significant distortions occur in the electric and magnetic fields at the material's surface, indicating the structural effects of the metamaterial. Strong electric and magnetic field intensities near the stepping and inside the honeycomb perforation suggest edge diffractions or secondary scattering in these areas. Thus, by combining gradient impedance multilayer structures with honeycomb perforations, the CuHT-FCIP-EP metamaterial achieves a broadband absorption from both material loss and structural effects. This mechanism effectively overcomes the skin effect faced by traditional single-layer EMW absorbers. As a representative technique in electronics, Smith Chart[40] provides a fast, less-computation, and graphical approach for evaluating the impedance matching performance. Analyzing the traces on the Smith Chart allows for a quick check to ensure that the absorber achieves the impedance at free space (the center of

the chart), thus achieving perfect absorption. As shown in the Fig. 4d–g, these phenomena result in a near-ideal impedance matching of the CuHT–FCIP–EP metamaterial, which is unattainable by single–component metamaterials. Therefore, a rational structural design allows the absorber to not only exhibit nearly ideal electromagnetic response but also simultaneously possess multiple loss mechanisms, thereby enhancing its EMW absorption capability.

## EMW absorption performance of CuHT–FCIP–EP metamaterial
To evaluate the EMW absorption performance of the CuHT–FCIP–EP metamaterial absorber, the metamaterial array was processed into a standard size (180 mm × 180 mm) plate sample and tested using the far-field method (Fig. 5b) with two separate horns (2–18 GHz and 18–40 GHz). The results show that the CuHT–FCIP–EP metamaterial absorber achieve effective EMW absorption across the entire 2–40 GHz frequency range (Fig. 5a). The number of resonance peaks in the experimental and simulated curves match well, with only partial deviations in peak positions, which presumably attributed to deviations in the electromagnetic properties between batches. Comparing with other recently reported 3D metamaterial absorbers, the

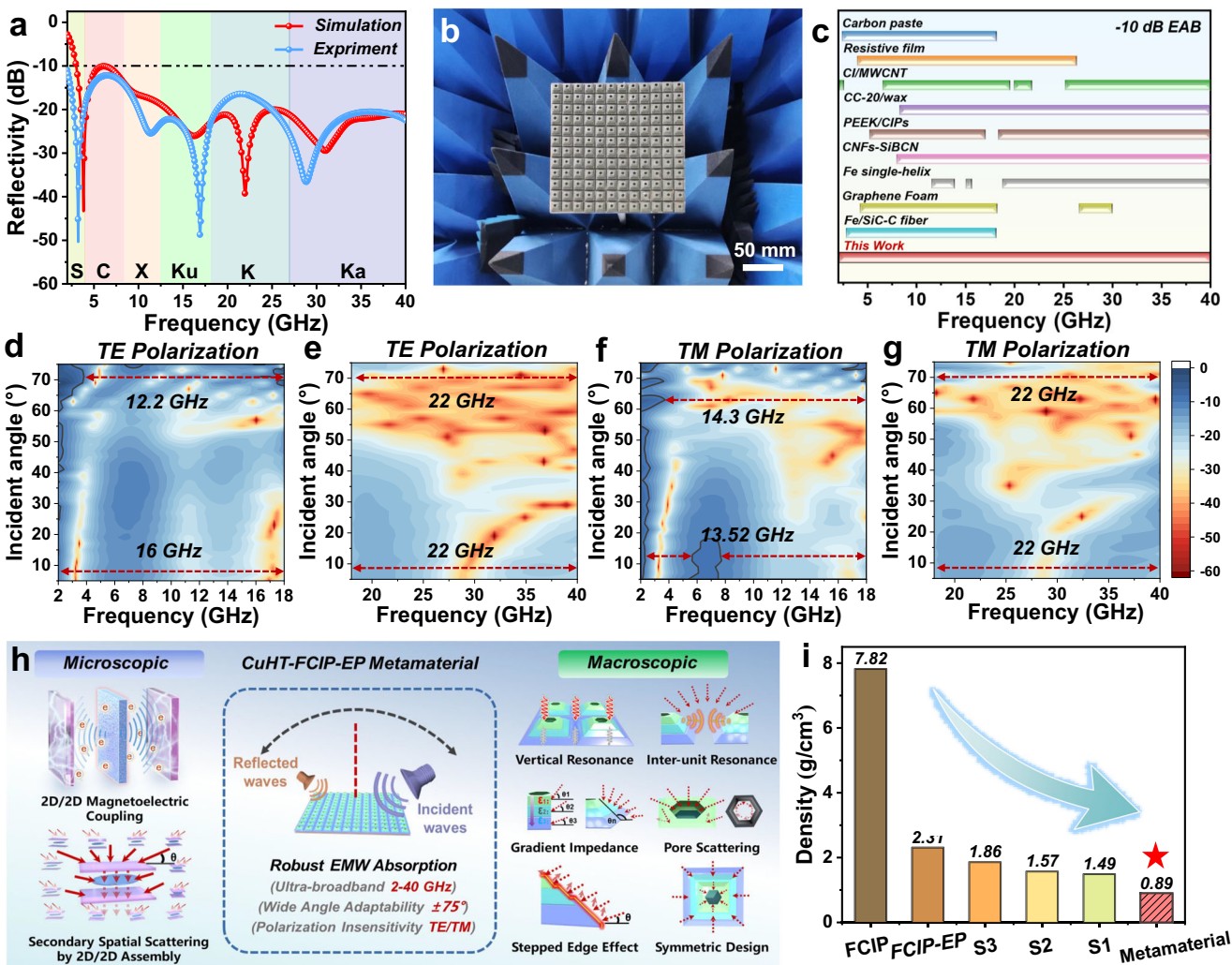

**Fig. 5 | EMW absorption performance. a** Comparison of experimental reflectivity and simulated reflectivity of ultra-broadband metamaterial absorber. **b** Photo of the CuHT-FCIP-EP metamaterial test sample. **c** Comparison of performance between CuHT-FCIP based metamaterial absorber and some typical 3D metamaterial. The relevant data are summarized in Supplementary Table 5. **d-g** Experimental reflectivity of metamaterial absorber at an oblique incidence with the incident angle ranging from 5° to 75° measured in both TE polarization and TM polarization. Source data are provided as a Source Data file. **h** Schematic illustration of the multi-scale absorption mechanism for CuHT-FCIP-EP metamaterial absorber. **i** Density performance of the proposed metamaterial absorber.

CuHT-FCIP-EP metamaterial absorber not only demonstrates an ultra-broad bandwidth (38 GHz) but also has a thinner thickness (9.3 mm, only 60% of the minimum thickness predicted by causality), presenting significant advancement (Fig. 5c). Furthermore, to the best of our knowledge, its −15 dB EAB (frequency ranges when RL < −15 dB), 35.85 GHz, is also found to be one of the broadest among the literatures (Supplementary Fig. 15).

In a complex electromagnetic environment, the performance of a microwave absorber must be averaged over various oblique incident angles under random polarizations. Hence it is of great importance to assess the robustness of the EMW absorption with different polarizations as well as with oblique incident angles. In the transverse electric (TE) polarization mode, the CuHT-FCIP-EP metamaterial absorber maintains consistent performance without significant deterioration up to 60°, achieving an EAB of 38 GHz in all cases. Even at an increased incident angle of 75°, it still retains an EAB of over 34.2 GHz (Fig. 5d, e). This excellent angle adaptability mainly stems from two factors: First, the frustum structure naturally inclines the metamaterial surface at approximately 18–30°, significantly reducing the real incidence angle with respect to the metamaterial sides; Second, the honeycomb perforations on the top surface and the spaces between units create

cavities that can induce additional secondary reflections at large incident angles, especially for shorter wavelength EMWs (Fig. 5h). This can be further confirmed by the significant decrease of RL at high-frequency ranges (18–40 GHz). In addition, owing to the C4 symmetry of the unit design, the metamaterial absorber exhibits polarization insensitivity. CuHT-FCIP-EP metamaterial demonstrates consistent and effective absorption performance in the transverse magnetic (TM) polarization mode across incident angles within 75°, with an EAB exceeding 35.52 GHz in all cases (Fig. 5f, g).

The mechanical performance of materials is equally important for electromagnetic metamaterials, especially in aerospace applications where specific strength is crucial. Metamaterials with higher specific strength not only provide additional structural support to prevent vibration damage but also enhance power economy. Despite the high density of FCIP, the CuHT-FCIP-EP metamaterial shows an impressively low density (0.89 g·cm⁻³) thanks to the inherently low-density CuHT (0.0856 g·cm⁻³) and EP (1.15 g·cm⁻³) (Fig. 5i). Cu-FCIP-EP composites have an impressive flexural modulus of up to 75 MPa. Therefore, Cu-FCIP-EP metamaterial demonstrates a specific compressive strength of up to 201.01 MPa m³ kg⁻¹, underscoring its exceptional mechanical strength performance (Supplementary Fig. 16).

## Discussion

We present a robust microwave absorber design based on the strong 2D/2D magnetoelectric coupling loss mechanism of semiconductor MOFs with ferromagnetic materials, as well as a three-dimensional metastructure design with gradient impedance arrangement. In previous works, metamaterial absorbers often work in the subwavelength regime, below a cutoff frequency determined by the lattice constant. Our work breaks this limit by introducing porous gradient structure design attendant with the strong magnetoelectric coupling absorbing material design, thereby effectively tuning the absorption bandwidth. The resulting CuHT-FCIP-EP metamaterial demonstrates near-perfect absorption from 2 GHz to 40 GHz, covering the entire 5 G high-frequency bands. Due to the introduction of magnetic materials, the thickness of the CuHT-FCIP-EP metamaterial is only 9.3 mm, yet it exhibits excellent absorption robustness (oblique angle within 75° and TE/TM polarization insensitivity). Despite the remarkable absorption performance, the straightforward manufacturing process and excellent mechanical performance make the CuHT-FCIP-EP metamaterial well-suited for the development and mass production of functional ultra-broadband EMW absorbers.

## Methods

### Materials

All the materials used in this work are provided in the supplementary file.

Synthesis of CuHT: CuHT was synthesized through a modified method based on a previous work[41]. Typically, 39.6 mmol 4-mercaptophenol (5.0 g) and 21.0 mmol cuprous oxide (3.0 g) were ultrasonically dissolved in 200 mL and 100 mL of anhydrous ethanol, respectively. The two solutions were then mixed into a 500 mL round-bottomed flask and reacted at 80 °C for 48 h in argon atmosphere. Afterward, the color of the solution changed from brick red to deep yellow. Subsequently, the precipitate was filtered and washed several times with anhydrous ethanol, N, N-dimethylformamide, deionized water, and acetone until greyish-yellow CuHT products were obtained with a yield of 85%.

Preparation of FCIPs: The spherical carbonyl iron powders (SCIPs) with an iron content of no less than 99.9% were sieved to obtain a desired size range (1–3 μm). Then, 316 stainless steel grinding balls (with diameters of 1, 2, 3, and 5 mm at a mass ratio of 1:6:2:1, respectively) were added to the mixing bowl of the mixer-mill in a ratio of 18:1 with respect to SCIP, along with an appropriate amount of anhydrous ethanol to facilitate dispersion. The ball-milling speed and time were set to 300 rpm and 10 h. After ball milling, the FCIPs were washed with acetone and dried in a vacuum oven at 50 °C for 12 h. Finally, the dried FCIPs were annealed in hydrogen atmosphere (20.95 sccm) for 6 h (350 °C, 5 °C·min⁻¹) to obtain the final FCIPs product.

Preparation of the CuHT-FCIP-EP composites: To prepare the CuHT-FCIP-EP composites, the CuHT-FCIP composites were first prepared. Typically, CuHT and FICPs with different mass fractions were mixed in an ethanol solution and then bath sonicated (40 W) in a nitrogen atmosphere at 60 °C for 6 h. Afterward, the treated solution was centrifuged (400 g, 10 min) and the precipitant was vacuum dried at 60 °C for 12 h to obtain the CuHT-FCIP products. Subsequently, CuHT-FCIP composites were weighed and mixed with epoxy resin (EP) at a mass ratio shown in Supplementary Table 1, and then degassed (0.1 bar) at 45 °C for 30 min. The mixed resin was then poured into the corresponding polytetrafluoroethylene mold for curing. The curing process included pre-curing at 50 °C for 3 h, curing at 80 °C for 5 h, and post-curing at 60 °C for 2 h at a heating/cooling rate of 1 °C·min⁻¹. The CuHT-EP, FCIP-EP and FICP-EP-H composites were prepared following the same procedures.

Preparation of the CuHT-FCIP-EP metamaterial: To prepare the CuHT-FCIP-EP metamaterial, a silicon rubber mold with specific features was first fabricated. The metamaterial was fabricated based on a multi-step molding process: (1) A specific amount of uncured CuHT-FCIP-EP-S1 resin was carefully poured into the silicone rubber mold to fill the bottom part of the mold (top layer of the metamaterial). (2) After curing for 2 h, the uncured CuHT-FCIP-EP-S2 resin was carefully poured into the silicone rubber mold to fill the middle layer. (3) After curing for another 2 h, the uncured CuHT-FCIP-EP-S3 resin was carefully poured into the silicone rubber mold to fill the rest cavities (bottom layer of the metamaterial). (4) Finally, after three additional hours of curing, CuHT-FCIP-EP metamaterials with S1–S3 combinations from top to the bottom were obtained by a complete demolding process.

Characterizations: The complex permittivity and permeability of all samples were measured using a vector network analyzer (VNA, M5463228, Anritsu, Japan) with the standard coaxial waveguide. The tested samples were fabricated using the corresponding mold with an outer diameter of 7.00 mm and an inner diameter of 3.04 mm. The RL values were calculated based on the transmission line theory according to the following equations[42]:

$$RL\ (\text{dB}) = 20 \log_{10} \frac{|Z-1|}{|Z+1|} \tag{1}$$

$$Z = \sqrt{\frac{\mu_r}{\varepsilon_r}} \tanh\left[ j \frac{2\pi f d}{c} \sqrt{\mu_r \varepsilon_r} \right] \tag{2}$$

where $\varepsilon_r$ and $\mu_r$ are the relative complex permittivity ($\varepsilon_r = \varepsilon' - j\varepsilon''$) and complex permeability ($\mu_r = \mu' - j\mu''$), respectively. $f$, $c$, $d$ denote the frequency of EMW, the speed of light, and the sample thickness, respectively, and represent the normalized input impedance of the tested sample. The dielectric loss ($\tan \varepsilon_r$) and magnetic loss factor ($\tan \mu_r$) are calculated according to the following equation:

$$\tan\varepsilon_r = \frac{\varepsilon''}{\varepsilon'} \quad \tan\mu_r = \frac{\mu''}{\mu'} \tag{3}$$

The far-field reflectivity (2–40 GHz) of the CuHT-FCIP-EP metamaterials was measured inside a microwave dark room. We adopted two pairs of double-ridge antennas, which were responsible for the frequency bands 2–18 GHz and 18–40 GHz. For a targeted frequency band, one antenna served as an emitter and the other as a receiver. The antennas were connected to the VNA via the coaxial cables whose cutoff frequency was higher than 40 GHz. A flat, 2.0-mm-thick copper plate with the same lateral dimensions as the sample was used to calibrate each measurement. We varied the polarizations of the incident microwave by rotating the azimuth of the antenna horn. The incidence angle is altered by controlling the position of the antenna fixed on the semi-circular rack. The measurement range of the incidence angle was 5°–75°, and the step size was set to 2° for data acquisition.

All remaining performance test methods or characterization details (material characterization, bandgap testing, mechanical testing, molecular and electromagnetic simulation techniques) are listed in the Supporting Information document.

## Data availability

The authors declare that all data supporting the findings of this study are available within the article and the Supplementary Information. All other data are available from the corresponding authors upon request. Source data are provided with this paper.

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

## Acknowledgements

The authors acknowledge the support from the National Science Fund for Distinguished Young Scholars (52025034, J.K.), National Natural Science Foundation of China (52203101, R.X.), National Natural Science Foundation of China (21875190, J.K.), National Key Research and Development Program of China (2022YFB3807101, J.K.), Innovation Team of Shaanxi Sanqin Scholars (J.K.), and the Innovation Foundation for Doctor Dissertation of Northwestern Polytechnical University (CX2024095, N.Q.).

## Author contributions

N.Q. and H.S. contributed equally to this work. J.K. supervised the project. N.Q., R.X., and J.K. conceived and designed the experiments. N.Q., H.S., Y.S., and M.H. performed experiments and acquired data. N.Q., R.X., J.G., and J.K. analyzed the data and drafted the manuscript. All authors supported the final formation of the manuscript.

## Competing interests

The authors declare no competing interests.
