## [Peer Review File · Nature Communications]

REVIEWER COMMENTS

Reviewer #1 (Remarks to the Author):

In this manuscript, the authors present a new electromagnetic metamaterial absorber design, which broad-spectrum absorption can span from 2 to 40 GHz. What makes this study particularly compelling is its reliance on precisely controlling the distribution of electromagnetic dissipative materials within a three-dimensional topological space. This involves intricate techniques such as assembling 2D/2D materials, dispersing assembly bodies, and superimposing different component assembly bodies. Besides, it can be seen that this metamaterial demonstrates excellent adaptability to oblique incidence and different polarization waves. I think that this point is an advancement compared to conventional patterned metamaterials. In short, this manuscript is well-organized, and the illustrations are both visually appealing and informative. This work is a strong contribution to the field of electromagnetic metamaterials, and I recommend it for publication. A few minor comments are included below:

(1) The proposed metamaterial design is very likely to draw the interest of researchers in the traditional metamaterials field. Could the authors please provide a detailed comparison in the main text, at an appropriate location, highlighting the differences and advantages (and possibly disadvantages) of this reported metamaterial in contrast to the traditional ones?

(2) In the work, the enhancement of impedance matching in metamaterial absorbers is demonstrated using a Smith chart. The authors should have included a more detailed explanation of the concepts related to Smith charts.

(3) Some of labels/legends in the pictures are small and vague, such as Figure 2l, m, please modify them.

(4) Please add unit to the axis (EAB and RL_min) in Figure 3i. Please indicate the EAB in Figure 5f, g. The caption of Figure 5i doesn't match the picture.

Reviewer #3 (Remarks to the Author):

The authors proposed a robust microwave absorber design based on the strong 2D/2D magnetoelectric coupling loss mechanism of semiconductor MOFs with ferromagnetic materials, as well as a three-dimensional metastructure design with gradient impedance arrangement. There are some questions should be addressed.

1. The author stated in the abstract that the purpose of the paper is to overcome the limitations of microwave absorbing materials in terms of bandwidth, low frequency, and robustness. The paper introduces the use of the 2D/2D magnetoelectric coupling loss mechanism of semiconductor

MOFs and three-dimensional metamaterial structures to achieve a -10 dB absorption bandwidth in the range of 2-40 GHz. The semiconductor material used in the fabrication process is simple and exhibits good robustness. However, the paper does not clearly explain the impact of the proposed method on low-frequency absorption. It is hoped that the authors can provide information on whether the proposed method can reduce low-frequency absorption frequencies and specify which part of the material is responsible for this effect.

2. The authors employed a three-layer gradient assembly of CuHT-FCIP-EP material, with each layer featuring a honeycomb perforation pattern. The choice of the honeycomb perforation pattern and the corresponding effects, as well as the rationale behind selecting the three-layer CuHT-FCIP-EP assembly, need to be explained by the authors. It is important to understand whether the 2-40 GHz absorption bandwidth is achieved through the three-dimensional gradient stacking of CuHT-FCIP-EP. It is also requested that the authors provide information on the absorption characteristics when using a single layer of CuHT-FCIP-EP with honeycomb perforations.

3. According to the Experimental section, the CuHT-FCIP-EP samples S1, S2, and S3 in the fabrication of the CuHT-FCIP-EP three-dimensional structure are arranged in the order from the bottom layer to the top layer, i.e., S1-S2-S3. However, in Figure 4a, the CuHT-FCIP-EP three-dimensional structure is depicted with the order reversed from the bottom layer to the top layer, i.e., S3-S2-S1. Please explain the impact of these two different three-dimensional arrangement sequences on the bandwidth. Furthermore, from Figure 4b, it can be observed that S1, S2, and S3 correspond to the absorption bandwidths for high frequency, mid-frequency, and low frequency, respectively. Please provide information on the combinations of different samples in the three-dimensional structure fabrication process and elaborate on the effects of different combinations on the absorption bandwidth, and need to give different sample absorption bandwidth division basis.

4. Figure 3i shows that the prepared CuHT-FCIP-EP is the best in the literature in terms of bandwidth, absorption and thickness. What are the advantages of CuHT-FCIP-EP 3D material in terms of bandwidth, absorption and thickness? Because Figure 5c only gives the bandwidth advantage of CuHT-FCIP-EP 3D material.

5. What is the thickness of samples S1, S2 and S3 in CuHT-FCIP-EP respectively? Because this thickness has a great influence on the absorption bandwidth in 3D CuHT-FCIP-EP. So what is the effect of the introduction of different thicknesses on the absorption bandwidth of 2D CuHT-FCIP-EP? Can the authors add a case of thickness effect in addition to examples of different CuHT/FCIP ratios?

6. What does "t FCIP" mean on page 10, line 15?

7. Last sentence on page 15 "Furthermore, to the best of our knowledge, its -15 dB EAB (frequency ranges when $RL < -15$ dB), 35.85 dB, is also found to be one of the trade-off among the three-month (Figure S12) ". What does it mean to have a -15db absorption bandwidth of 35.85dB?

The title of the article is "2D/2D Coupled MOF/Fe Composite Metamaterials Enable Robust Ultra-Broadband Microwave Absorption", the authors have realized microwave absorbing materials with a wide absorption bandwidth by constructing two-dimensional MOF/Fe composite structures. The material has good absorption properties, and the testing method can explain the absorption mechanism. However, it still needs some modifications as well as explanations before it can be accepted.

1. The microwave absorption properties of the composites in the frequency range of 2-40 GHz are given in Figure S9, can the authors add pictures of the data such as permittivity and permeability for this section?
2. It is well known that carbon materials and carbon matrix composites have a very important place in broadband microwave absorption. such as 1. Multiple synergistic losses in the absorption of electromagnetic waves by three-dimensional cross-linked carbon fiber. Carbon, 2022, 195, 308-318. 2. The lightweight electromagnetic wave absorbing composites with Fe₃O₄ nanocrystals uniformly decorated on the surface of carbon spheres. Nanoscale, 2022, 14(29), 10456-10468. The authors are asked to describe what advantages the composites prepared in this manuscript have over carbon matrix composites.
3. The authors are requested to describe in detail the process of testing the wave absorption properties. Whether the wave absorption properties of composites in the frequency range of 2-40 GHz are tested in segments or directly can be tested for wave absorption in that frequency range.

Reviewer #1:

In this manuscript, the authors present a new electromagnetic metamaterial absorber design, which broad-spectrum absorption can span from 2 to 40 GHz. What makes this study particularly compelling is its reliance on precisely controlling the distribution of electromagnetic dissipative materials within a three-dimensional topological space. This involves intricate techniques such as assembling 2D/2D materials, dispersing assembly bodies, and superimposing different component assembly bodies. Besides, it can be seen that this metamaterial demonstrates excellent adaptability to oblique incidence and different polarization waves. I think that this point is an advancement compared to conventional patterned metamaterials. In short, this manuscript is well-organized, and the illustrations are both visually appealing and informative. This work is a strong contribution to the field of electromagnetic metamaterials, and I recommend it for publication. A few minor comments are included below:

Response: Thank you very much for your thoughtful comments. We are truly pleased to receive your positive feedback and your recognition of the contributions and potential significance of our research. The comments and concerns are addressed point by point in the following text.

(1) The proposed metamaterial design is very likely to draw the interest of researchers in the traditional metamaterials field. Could the authors please provide a detailed comparison in the main text, at an

appropriate location, highlighting the differences and advantages (and possibly disadvantages) of this reported metamaterial in contrast to the traditional ones?

Response: We greatly appreciate the reviewers' suggestion. This work presents a novel design of microwave metamaterial absorber, whose absorption mechanism differs from the traditional metamaterial absorbers. As a result, it achieves an ultra-broadband, ultra-wide-angle, and polarization-insensitive microwave absorption performance. The specific differences and advantages compared to traditional metamaterials are as follows:

Traditional metamaterial absorbers use conductive patterns to achieve ideal impedance matching of electromagnetic waves at a certain frequency through the electric resonance between metallic conductors. Despite recent efforts by researchers to enhance absorption bandwidth using techniques such as incorporating resistive sheets or loading with lumped elements, or superposing resonant units in circuits, traditional metamaterials still suffer from limitations in the absorption bandwidth, particularly in the microwave frequency range of 2-40 GHz.

The metamaterial in this paper mainly relies on the dielectric loss of electromagnetic waves in three-dimensional materials and the secondary reflection loss formed within specific topological structures, thus exhibiting improved absorption bandwidth and adaptability to incident angles compared to the traditional materials. Such metamaterials can be treated as a class of variants of traditional metamaterials, also known as '3D metamaterials'. However, the performance of 3D metamaterials relies heavily on the electromagnetic loss properties of the materials themselves, with the most optimal performance typically include complex micro/nanostructures such as multi-core-shell nanoparticles, nanoflowers and hierarchical nanoparticles. This greatly compromise the stability of magnetoelectric coupling of materials and, hence, the robustness (i.e., oblique, and polarized incidence) of absorption for both experimental implementation and mass production.

Our study employs a 2D/2D material assembly strategy to tackle the aforementioned issues in 3D metamaterials. We discovered the strong magnetoelectric coupling effect between MOF crystal materials and carbonyl iron materials, which enables effective modulation of magnetoelectric coupling across various frequency ranges. Through further topological design, we have achieved ultra-broadband and highly robust microwave absorption effects. Compared to traditional metamaterials, the performance of the metamaterials we report can be tuned not only by structure design but also by manipulating the

properties of the absorbing filler. This represents a comprehensive outcome of multiscale design, spanning various levels from atomic and molecular to crystal and macroscopic structures.

The corresponding discussions are updated in the main text and are highlighted in red:

Page 3

“The recently emerged three–dimensional (3D) metamaterials, leveraging the 3D topological structure, have showcased promising bandwidth by further incorporating dielectric/magnetic losses and diffraction resonances.⁸ However, the performance of 3D metamaterials relies heavily on the EMW loss properties of the materials themselves⁹, with the most optimal performance typically include complex micro/nanostructures (e.g., multi-core-shell nanoparticles¹⁰, nanoflowers¹¹ and hierarchical nanoparticles¹²). This greatly compromise the stability of magnetoelectric coupling of materials and, hence, the robustness (i.e., oblique, and polarized incidence) of absorption for both experimental implementation and mass production.”

Page 20

“In previous works, metamaterial absorbers often work in the subwavelength regime, below a cutoff frequency determined by the lattice constant. Our work breaks this limit by introducing porous gradient structure design attendant with the strong magnetoelectric coupling absorbing material design, thereby effectively tuning the absorption bandwidth.”

(2) In the work, the enhancement of impedance matching in metamaterial absorbers is demonstrated using a Smith chart. The authors should have included a more detailed explanation of the concepts related to Smith charts.

Response: We greatly appreciate the suggestion. We have added the concepts related to Smith charts accordingly. The corresponding revisions are highlighted in red and referred as bellow:

Page 15

“Smith Chart⁴² provides a fast, less-computation, and graphical approach for evaluating the impedance matching performance. Analyzing the traces on the Smith Chart allows for a quick check to ensure that the absorber achieves the impedance at free space (the center of the chart), thus achieving perfect absorption.”

(3) Some of labels/legends in the pictures are small and vague, such as Figure 2l, m, please modify them.

Response: Thanks for the feedback. We have modified the size of the labels in Figure 2l, m and have replaced them in the manuscript. The revised figures are referred as below:

Figure 2 in the revised manuscript

(4) Please add unit to the axis (EAB and RL_{min}) in Figure 3i. Please indicate the EAB in Figure 5f, g. The caption of Figure 5i doesn't match the picture.

Response: Thanks for your feedback. We sincerely apologize for any confusion. We have reviewed all the figures in the manuscript and have addressed the identified issues. The updated figures are provided below for your reference:

Figure 3 in the revised manuscript

Figure 5 in the revised manuscript

Figure 5i caption: "Density performance of the proposed metamaterial absorber".

Reviewer #2:

The title of the article is "2D/2D Coupled MOF/Fe Composite Metamaterials Enable Robust Ultra-Broadband Microwave Absorption", the authors have realized microwave absorbing materials with a wide absorption bandwidth by constructing two-dimensional MOF/Fe composite structures. The material has good absorption properties, and the testing method can explain the absorption mechanism. However, it still needs some modifications as well as explanations before it can be accepted.

Response: We greatly appreciate your kind words and acknowledgment of the value of our research. Your positive feedback is truly encouraging and motivates us to continue our work with dedication. The comments and concerns are addressed point by point in the following text.

1. The microwave absorption properties of the composites in the frequency range of 2-40 GHz are given in Figure S9, can the authors add pictures of the data such as permittivity and permeability for this

section?

Response: Thanks for your comment. We sincerely apologize for any misunderstanding caused to the reviewers. The absorption curves in Figure S9 (**Figure S11** in the revised SI) represent the simulated reflectance curves of the metamaterials using finite element method (CST STUDIO SUITE 2022). We utilized dielectric and magnetic parameters from CuHT-FCIP-EP-S1, CuHT-FCIP-EP-S2, and CuHT-FCIP-EP-S3, as depicted in **Figures S7a-e**. These parameters were arranged in the same sequence as in **Figure 4a** and employed as inputs for a frequency domain solver to obtain the reflection loss curves presented in Figure S9 (**Figure S11** in the revised SI).

During the optimization process of the proposed metamaterial structure parameters, we defined ranges and search steps for different structural parameters (T1-T5, L1-L7) and utilized optimization solvers to obtain optimal structural parameters (**Table S4**). To succinctly illustrate the trends of each structural parameter on the metamaterial performance, we selected five representative values for each factor within the specified range and plotted them in **Figure S11**.

Therefore, to avoid misunderstandings from the readers, we have added additional comments in the SI: “Based on the permittivity and permeability of CuHT-FCIP-EP-S1, CuHT-FCIP-EP-S2, and CuHT-FCIP-EP-S3 (Figure S6), we calculated the reflection loss curves of the multilayered composite metamaterials with different geometry parameters by CST STUDIO SUITE 2022. Floquet boundary conditions were used in this process, the frequency range was set to 2-40 GHz, and the structure was meshed in a hexahedral form.”

2. It is well known that carbon materials and carbon matrix composites have a very important place in broadband microwave absorption. such as 1. Multiple synergistic losses in the absorption of electromagnetic waves by three-dimensional cross-linked carbon fiber. Carbon, 2022, 195, 308-318. 2. The lightweight electromagnetic wave absorbing composites with Fe₃O₄ nanocrystals uniformly decorated on the surface of carbon spheres. Nanoscale, 2022, 14, 10456-10468. The authors are asked to describe what advantages the composites prepared in this manuscript have over carbon matrix composites.

Response: Thanks for the comments. Indeed, carbon materials are popular choices for EMW absorption due to their excellent electrical conductivity, thereby possessing strong electromagnetic loss capabilities. Thanks to the advancements in catalyst and energy research, doping carbon materials with metals or non-

metal components can further enhance their electromagnetic loss mechanisms, leading to improved absorption performance. However, the fabrication of most carbon materials typically involves precursor materials undergoing high-temperature carbonization processes, introducing uncertainties into the atomic-level structure of the materials. For example, carbon materials derived from MOF precursors may undergo potential framework collapse and phase transitions of metallic elements during pyrolysis, leading to batch-to-batch performance instability—an aspect often overlooked in many studies. In contrast, the synthesis process of conductive MOFs in this study involves entropy-driven crystallization, offering highly controllable characteristics. In addition, the high-energy consumption associated with high-temperature pyrolysis also contradicts the precise and environmentally friendly synthesis principles of future materials processing.

Besides, traditional methods of combining carbon materials with magnetic materials typically involve either doping magnetic components into the carbon matrix or direct mixing. The former often results in insufficient magnetic properties of the materials, with impedance matching and low-frequency performance of carbon materials remaining inadequate. whereas the latter can lead to the isolation of magnetic components by carbon materials, resulting in the loss of magnetic domain coupling between magnetic components, thereby reducing the magnetic loss performance of the materials. The 2D semiconductive MOF (CuHT) prepared in this study feature abundant thiol functional groups on their surfaces, which can effectively bond with magnetic transition metals (e.g., Fe), achieving layer-by-layer assembly. As a result, effective magneto-electric coupling loss are formed between the added magnetic components and dielectric components, ultimately imparting excellent electromagnetic loss performance to the materials in terms of EAB, RL_{\min} , and thickness (**Figure 3i**).

In summary, the 2D/2D composite absorber (CuHT-FCIP) presented in this study provides an alternative preparation method that does not involve pyrolysis, unlike traditional carbon materials, particularly those derived from MOF precursors. It features environmentally friendly and stable processing. Through the covalent assembly of CuHT and FCIP, efficient magneto-electric coupling loss is achieved, demonstrating excellent intrinsic EAB, RL_{\min} , and low thickness.

Page 12

“Compared with other recently reported MOF-derived carbon materials, CuHT-FCIP-EP demonstrates

unparalleled advantages in terms of EAB, thickness, and maximum absorption intensity (Figure 3i). More importantly, it offers a pyrolysis-free preparation process, distinguishing it further.”

3. The authors are requested to describe in detail the process of testing the wave absorption properties. Whether the wave absorption properties of composites in the frequency range of 2-40 GHz are tested in segments or directly can be tested for wave absorption in that frequency range.

Response: Thanks for the feedback. The reflectivity of the metamaterial is measured using the far-field methods. The data within the 2-40 GHz range were collected separately, divided into two segments: 2-18 GHz and 18-40 GHz (as depicted in **Figure 5d-g**). To address the reviewers’ concern, we modified the corresponding words for clarity:

Page 16

“To evaluate the EMW absorption performance of the CuHT-FCIP-EP metamaterial absorber, the metamaterial array was processed into a standard size (180 mm×180 mm) plate sample and tested using the far-field method (Figure 5b) with two separate horns (responsible for 2-18 GHz and 18-40 GHz).”

Page S22 (Experimental section)

“The far-field reflectivity (2-40 GHz) of the CuHT-FCIP-EP metamaterials was measured inside a microwave dark room. We adopted two pairs of double-ridge antennas, which were responsible for the frequency bands 2-18 GHz and 18-40 GHz.”

Reviewer #3 (Remarks to the Author):

The authors proposed a robust microwave absorber design based on the strong 2D/2D magnetoelectric coupling loss mechanism of semiconductor MOFs with ferromagnetic materials, as well as a three-dimensional metastructure design with gradient impedance arrangement. There are some questions should be addressed.

Response: We greatly appreciate your in-depth comments and suggestions, which have greatly improved the manuscript. The concerns are addressed point-by-point as stated below:

1. The author stated in the abstract that the purpose of the paper is to overcome the limitations of microwave absorbing materials in terms of bandwidth, low frequency, and robustness. The paper

introduces the use of the 2D/2D magnetoelectric coupling loss mechanism of semiconductor MOFs and three-dimensional metamaterial structures to achieve a -10 dB absorption bandwidth in the range of 2-40 GHz. The semiconductor material used in the fabrication process is simple and exhibits good robustness. However, the paper does not clearly explain the impact of the proposed method on low-frequency absorption. It is hoped that the authors can provide information on whether the proposed method can reduce low-frequency absorption frequencies and specify which part of the material is responsible for this effect.

Response: Thanks for the suggestion. We apologize for the lack of clarity in explaining the low-frequency (2-5 GHz) absorption mechanism in the previous manuscript. In this work, we believe that the outstanding low-frequency performance of CuHT-FCIP-EP metamaterials stems from the synergistic effects of the following factors, including (1) the introduction of strong magnetic component (FCIP), (2) the construction of the magnetoelectric coupling network in CuHT-FCIP, and (3) the rational design of a multilayer 3D structure.

First, the FCIP used in this work exhibits outstanding magnetization strength (145.9 emu/g). Moreover, FCIP's two-dimensional structure allows it to exceed the Snoek limit, resulting in a higher initial magnetization. Additionally, under low-frequency conditions, magnetic materials are more susceptible to magnetization and show a more pronounced response to external magnetic fields. This magnetization response facilitates additional energy dissipation (DOI: 10.1002/adfm.202213258).

Second, low-frequency EMW absorption relies on both dielectric and magnetic losses in materials. The 2D/2D assembly formed by CuHT and FCIP creates an efficient magneto-electric coupling network. On one hand, the two-dimensional semiconductor MOF, CuHT, facilitates electron transfer between layers (**Figure 2h-k**). On the other hand, the interlayer structure of CuHT-FCIP avoids magnetic components being separated by dielectric components, preventing magnetic decoupling. Therefore, the CuHT-FCIP composite in this study exhibits excellent absorption performance at low frequencies. For example, CuHT-FCIP-EP-S3 achieves absorption performance of -37 dB (99.98% absorption) at 2.92 GHz; CuHT-FCIP-EP-S2 also achieves absorption performance of -61 dB (99.9999% absorption) at 5.2 GHz. This indicates that CuHT-FCIP itself possesses excellent low-frequency absorption capability.

Third, the structural design of the metamaterial further enhances its absorption capacity in the low-frequency region. As shown in **Figure S13**, the main resonance absorption peaks of CuHT-FCIP-EP-S3,

S2, and S1 occur at 5.02 GHz, 7.51 GHz, and 15.4 GHz, respectively, indicating that the low-frequency absorption of the metamaterial mainly relies on the S3 material, i.e., the bottom layer structure. Additionally, the perforated structure of the metamaterial promotes the entry and multiple reflections of EMWs into the material's cavity, effectively mitigating the skin effect caused by the increase in dielectric constant and thickness of the material, thus further enhancing the EMW absorption performance across the entire frequency range. Furthermore, at low frequency range, the simulated results of electromagnetic field distribution in **Figure 4c** show significant enrichment at the bottom part of the metamaterial, reechoing the contribution of the bottom layer material to the absorption of low-frequency EMWs.

In summary, we believe that the strong magnetoelectric network construction of CuHT-FCIP, along with the design of the three-layer slab and honeycomb perforated structure in the metamaterial, collectively contribute to the outstanding low-frequency absorption performance of the CuHT-FCIP-EP metamaterial.

To better demonstrate this fact, we have made the following revisions:

Page 15

“Specifically, the peaks at 16.1 GHz and 32.2 GHz are mainly attributed to thickness resonances at $\lambda/2$ and λ (wavelength), while the peak at 3.82 GHz and 22.0 GHz are primarily attributed to the resonance of the bottom layer (S3) and the honeycomb structure.³⁹ (Figure 5a, Figure S13). The honeycomb perforation enables increased entry of low-frequency EMWs. Consequently, at 3.82 GHz, the electric and magnetic fields exhibit a concentrated distribution in the bottom layer of the metamaterial. This suggests that the bottom layer material (i.e., CuHT-FCIP-EP-S3) primarily functions in absorbing EMWs at the low-frequency range (Figure S14). As the wavelength decreases, significant distortions occur in the electric and magnetic fields at the material's surface, indicating the structural effects of the metamaterial. Strong electric and magnetic field intensities near the stepping and inside the honeycomb perforation suggest edge diffractions or secondary scattering in these areas. Thus, by combining gradient impedance multilayer structures with honeycomb perforations, the CuHT-FCIP-EP metamaterial achieves a broadband absorption from both material loss and structural effects. This mechanism effectively overcomes the skin effect faced by traditional single-layer EMW absorbers.”

SI (Figure S13)

Figure S13. Simulated EMW absorption performance of the proposed metamaterial structure. (a) Bottom layer (S3), (b) Middle layer (S2), (c) Top layer (S1), (d) Multi-layer structure (top to bottom: S1-S2-S3) and (e) Multi-layer structure (top to bottom: S1-S2-S3) with a honeycomb perforation. (f) Basic rationale for the effectiveness of multilayer design.

“Building upon this, a honeycomb perforation is incorporated on top of the metamaterial to expand the EAB further by introducing structural resonances, which effectively mitigates the skin effect caused by the increase in dielectric constant and thickness of the material. As a result, another two distinct absorption peaks appear (3.82 GHz and 22.0 GHz) while the former two resonance peaks remain unchanged. (Figure S13e-f) This confirms the uniqueness and validity of our proposed novel metamaterial model.”

Furthermore, in order to emphasize the superiority of the low-frequency range absorption performance, we added the comparison of EAB in S-C band (i.e., 2-8 GHz) in **Table S5**. The revised table is referred in the response of Question 4.

2. The authors employed a three-layer gradient assembly of CuHT-FCIP-EP material, with each layer featuring a honeycomb perforation pattern. The choice of the honeycomb perforation pattern and the corresponding effects, as well as the rationale behind selecting the three-layer CuHT-FCIP-EP assembly, need to be explained by the authors. It is important to understand whether the 2-40 GHz absorption bandwidth is achieved through the three-dimensional gradient stacking of CuHT-FCIP-EP. It is also requested that the authors provide information on the absorption characteristics when using a single

layer of CuHT-FCIP-EP with honeycomb perforations.

Response: Thanks for the feedback. The enhancement of the absorption bandwidth for the CuHT-FCIP-EP metamaterial due to honeycomb perforations can be observed from **Figures S13d-e**. It is evident that after honeycomb perforation, two distinct resonance peaks appear on the absorption curve at 3.82 GHz and 22.0 GHz, which correspond to the resonance peak of the bottom layer and the honeycomb structure (**Figure S13a**). The EAB has also been further extended from 5.2-40 GHz to 2-40 GHz. This indicates a significant enhancement of the absorption capability in the low-frequency region upon introducing honeycomb perforations. The primary reason is that after honeycomb perforations, low-frequency EMWs (with longer wavelengths) can more easily enter the holes, thereby undergoing dissipation through the bottom layer material (S3 and partial S2) below the holes. This point is supported by the simulation results of the distribution of electric fields, magnetic fields, and power loss density in the metamaterial at low frequencies (detailedly stated in Question 1). Additionally, from **Figure S13f** (referred below), it can be observed that the depth of the honeycomb perforations is 4.4 mm, leading to the thickness of the material below the honeycomb perforation being 4.9 mm. This precisely corresponds to the resonance thickness (4.9 mm) of CuHT-FCIP-EP-S3 at 2.92 GHz of the metal backplane model (**Figure 3c**), reechoing the mechanism of low-frequency absorption.

Figure S13f.

Besides, the hexagonal honeycomb perforation structure allows more entrance of the high-frequency EMWs into the cavity and induces the edge resonance effect of high-frequency EMWs. As shown in **Figure 4c** and **Figure S12**, in the high-frequency region, the open edges of the honeycomb perforations exhibit strong concentration of electromagnetic field losses. This also explains the appearance of the additional absorption peak at 22 GHz in **Figure S13e**.

The hexagonal honeycomb perforation was chosen as the optimal structure for the following two reasons:

(1) The hexagonal honeycomb structure is a kind of geometric structure with well-documented electromagnetic resonance effects (DOI: 10.1039/d0cs00320d, 10.1021/acsnano.1c01552, 10.1016/j.carbon.2020.05.045). It possesses good symmetry (C6 symmetry) and spatial utilization efficiency. Based on the analysis above, the primary purpose of introducing perforations is to facilitate the entrance of low-frequency (with longer wavelength) EMWs into the perforation and inducing edge resonance of high-frequency EMWs. Therefore, selecting a geometric structure with the maximum cross-sectional area and retaining as many characteristic straight edges as possible is desired. To achieve this, we simulated the reflection loss curves of perforated structures with different numbers of side lengths (as shown in **Figure S14**). The results show that as the number of edges of the perforation structure increases, the characteristic absorption peak in the low-frequency region gradually approaches 2 GHz. This indicates that increasing the cross-sectional area facilitates the entrance of low-frequency EMWs and further causes losses. However, as the number of edges increases, the length of the edge decreases (for a circle, the side length can be considered infinitely small), thereby weakening the edge resonance effect of high-frequency EMWs. Therefore, ultimately, we choose the hexagonal perforation structure.

In response to the reviewer's request, we have provided simulation results of the reflection loss for a single-layer structure with perforations having the same parameters as the optimal structure (**Figure S14e-g**). Comparing with the single-layer structure without perforations (**Figure S13a-c**), it can be observed that the honeycomb perforations do not lead to significant changes in the position or intensity of absorption peaks for single layer structures. Therefore, for a single-layer structure, the enhancement of absorption performance due to honeycomb perforations is limited. This indirectly demonstrates the crucial role of perforations in improving low-frequency EMW absorption performance and inducing high-frequency resonance effects in a multilayer structure.

(2) Honeycomb structures are widely acknowledged in engineering for their exceptional mechanical load-bearing capabilities. This is attributed to their inherent ability to offer higher stiffness and strength with minimal weight. The design of honeycomb structures, inspired by nature, enables them to distribute loads uniformly in all directions while utilizing minimal material. As a result, honeycomb structures find

extensive applications in aerospace, vehicle manufacturing, and various other fields that demand lightweight yet robust solutions [DOI: 10.1021/acsomega.3c10138]. In this work, with the help of honeycomb perforation, the bulk density of the metamaterial is only 0.89 g/cm³ leading to an excellent specific compressive strength (201.01 MPa·m³·kg⁻¹).

The revisions are listed below for your reference:

Page 15

“Specifically, the peaks at 16.1 GHz and 32.2 GHz are mainly attributed to thickness resonances at $\lambda/2$ and λ (wavelength), while the peak at 3.82 GHz and 22.0 GHz are primarily attributed to the resonance of the bottom layer (S3) and the honeycomb structure.³⁹ (Figure 5a, Figure S13). The honeycomb perforation enables increased entry of low-frequency EMWs. Consequently, at 3.82 GHz, the electric and magnetic fields exhibit a concentrated distribution in the bottom layer of the metamaterial. This suggests that the bottom layer material (i.e., CuHT-FCIP-EP-S3) primarily functions in absorbing EMWs at the low-frequency range (Figure S14). As the wavelength decreases, significant distortions occur in the electric and magnetic fields at the material's surface, indicating the structural effects of the metamaterial. Strong electric and magnetic field intensities near the stepping and inside the honeycomb perforation suggest edge diffractions or secondary scattering in these areas. Thus, by combining gradient impedance multilayer structures with honeycomb perforations, the CuHT-FCIP-EP metamaterial achieves a broadband absorption from both material loss and structural effects. This mechanism effectively overcomes the skin effect faced by traditional single-layer EMW absorbers.”

SI (Figure S14)

Figure S14. Simulation of reflection loss for metamaterials with different geometric perforation features. (a) Square, (b) Hexagon, (c) Octagon and d Cylinder. (e-g) Single-layer hexagonal honeycomb perforation.

3. According to the Experimental section, the CuHT-FCIP-EP samples S1, S2, and S3 in the fabrication of the CuHT-FCIP-EP three-dimensional structure are arranged in the order from the bottom layer to the top layer, i.e., S1-S2-S3. However, in Figure 4a, the CuHT-FCIP-EP three-dimensional structure is depicted with the order reversed from the bottom layer to the top layer, i.e., S3-S2-S1. Please explain the impact of these two different three-dimensional arrangement sequences on the bandwidth. Furthermore, from Figure 4b, it can be observed that S1, S2, and S3 correspond to the absorption bandwidths for high frequency, mid-frequency, and low frequency, respectively. Please provide information on the combinations of different samples in the three-dimensional structure fabrication process and elaborate on the effects of different combinations on the absorption bandwidth, and need to give different sample absorption bandwidth division basis.

Response: Thanks for the feedback and great suggestion. We apologize for any confusion regarding the description of the layer order in the multilayer structure. In this work, we adopted a stacking order from top to bottom, namely S1-S2-S3, for assembling the multilayer metamaterial (as shown in **Figure 4a**). We made this selection because, in this sequence, the three layers demonstrate the most effective power loss coverage across low, mid, and high frequencies, as illustrated in **Figure 4b**. To make it more clear, we modified the experimental section:

Page 21

“Preparation of the CuHT–FCIP–EP metamaterial: To prepare the CuHT–FCIP–EP metamaterial, a silicon rubber mold with specific features was first fabricated. The metamaterial was fabricated based on a multi–step molding process: 1) A specific amount of uncured CuHT–FCIP–EP–S1 resin was carefully poured into the silicone rubber mold to fill the bottom part of the mold (top layer of the metamaterial). 2) After curing for 2 h, the uncured CuHT–FCIP–EP–S2 resin was carefully poured into the silicone rubber mold to fill the middle layer. 3) After curing for another 2 h, the uncured CuHT–FCIP–EP–S3 resin was carefully poured into the silicone rubber mold to fill the rest cavities (bottom layer of the metamaterial). 4) Finally, after three additional hours of curing, CuHT-FCIP-EP metamaterials with S1-S3 combinations from top to the bottom were obtained by a complete demolding process.”

As for the sequence of the three layer, we determined the sequence through simulation optimization, and the specific results can be found in **Figure S10** (referred below).

Figure S10. Effect of different material combinations on the bandwidth of metamaterials. The simulated reflectivities of metamaterial absorbers are composed of different sequences listed in Table S4 in the frequency range of 2-40 GHz.

Table S3. CuHT-FCIP-EP metamaterials based on different material combinations.

ID	R1	R2	R3	R4	R5	R6
Top-layer	S1	S2	S1	S3	S3	S2
Middle-layer	S2	S1	S3	S1	S2	S3
Bottom-layer	S3	S3	S2	S2	S1	S1

Corresponding discussions are added in the revised manuscript:

Page 13

“The results reveal that when arranged from top to bottom as S1–S2–S3, the power loss contributions of the overall material perfectly match and connect across the low–frequency range (2-6 GHz), mid–

frequency range (6-18 GHz), and high-frequency range (18-40 GHz). (**Figure 4b**) This corresponds to the characteristic impedance matching frequency of each layer material in Figure 3g and eventually ensures a broadband absorption across the entire 2-40 GHz range (**Figure S10, Table S3**).”

SI (Figure S10)

“All possible three-layer combinations are detailed in **Table S3**. Comparing the simulated reflectivity results between R1, R2, and other combinations, it is evident that when S3 is placed at the bottom layer, the metamaterial exhibits superior absorption performance in the low-frequency range (2-5 GHz). This can be attributed to the higher content of strong magnetic component in S3, which maximizes the absorption of low-frequency EMWs. Further comparison between R1 and R2 results shows that, despite similar -10 dB EAB, when S1 is the top layer, the resonance absorption peak intensities of the metamaterial are significantly higher compared to the scenario when S2 is at the top. This is because S1 material has relatively lower attenuation constant, resulting in a gradient distribution from top to bottom. Such a distribution has been widely acknowledged to enhance the efficiency of electromagnetic absorption. Consequently, the combination of S1-S2-S3 was chosen as the optimal configuration.”

4. Figure 3i shows that the prepared CuHT-FCIP-EP is the best in the literature in terms of bandwidth, absorption and thickness. What are the advantages of CuHT-FCIP-EP 3D material in terms of bandwidth, absorption and thickness? Because Figure 5c only gives the bandwidth advantage of CuHT-FCIP-EP 3D material.

Response: Thank you for the feedback. In **Table S5**, we provide a detailed comparison of the technical parameters of the 3D metamaterial reported in this paper with the recent literature, including -10 dB and -15 dB EAB, as well as -10 dB EAB in the low-frequency range (S-C band, 2-5 GHz), thickness, minimum reflection loss (RL_{\min}), compression strength, etc.

First, compared to recent reports on metamaterials with three-dimensional structures, the CuHT-FCIP-EP metamaterial exhibits outstanding performance in the bandwidth range of 2-40 GHz. Not only does it achieve full -10 dB coverage (90% absorption) from 2 to 40 GHz, but its -15 dB EAB (97% absorption) also reaches 35.85 GHz, covering 94.3% of the whole bandwidth. Besides, the metamaterial also possesses remarkably robust absorption performance, manifested in its wide-angle adaptability from 4° to 75° and insensitivity to TE/TM polarization.

Second, the CuHT-FCIP-EP metamaterial achieves a RL_{\min} of -51 dB, and at least in three frequency positions (2.6 GHz, 17.1 GHz, and 28.2 GHz), the RL_{\min} are all greater than -35 dB (99.97% absorption). Meanwhile, its average reflection loss in the range of 2-40 GHz is -22.23 dB (99.40% absorption), indicating significant advantages of the CuHT-FCIP-EP metamaterial in both targeted frequency absorption and broadband absorption.

Third, the thickness of the CuHT-FCIP-EP metamaterial is only 9.30 mm, which is smaller than the majority of current 3D metamaterials. Moreover, for the low-frequency bands (2-5 GHz) that typically require thicker materials for effective absorption, our metamaterial achieves full-frequency absorption at significantly less than 1/4 of the wavelength.

Fourth, current reports on 3D metamaterials often overlook the mechanical properties of the structures. The CuHT-FCIP-EP metamaterial we report employs structurally stable design (truncated pyramid) and epoxy resin as a molding substrate, exhibiting excellent compressive strength (201.01 MPa). Additionally, due to the low density of CuHT, the overall density of CuHT-FCIP-EP metamaterial is only 0.89 g/cm^3 , making it also lightweight yet high-strength.

In short, the CuHT-FCIP-EP metamaterial offers significant advantages not only in bandwidth and thickness, but also in absorption robustness and mechanical strength.

The corresponding revisions are made to improve clarity:

Page 16

“Comparing with other recently reported 3D metamaterial absorbers, the CuHT–FCIP–EP metamaterial absorber not only demonstrates an ultra–broad bandwidth (38 GHz) but also has a thinner thickness (9.3 mm, only 60% of the minimum thickness predicted by causality), presenting significant advancement (Figure 5c).”

Figure 5c caption

“The relevant data are summarized in Table S5.”

SI (Table S5)

Table S5.

Sample	Component	EAB (GHz) <-10 dB	EAB (GHz) <-15 dB	EAB (<- 10 dB) (S-C band)	Thickness (mm)	RL _{min} (dB)	Compr. Strength (MPa)	Den sity Test	Ref.
Carbon coated corrugated structure	Carbon paste	2.31-18.00	3.00-5.00, 9.00-12.50, 16.20-18.00	5.69	18.00	-25.93	No	No	S25
3D resistance film structure	Resistive film	3.90-26.20	20.80-22.00	4.10	11.00	-15.90	No	No	S26
Spherical CI/MWCNT epoxy resin	CI/MWCNT	2.00-2.36, 6.54-19.36, 19.89-21.57, 25.10-40.00	9.78-18.00, 27.14-36.11	1.82	7.00	-54.00	No	No	S27
3D printed shell with CC-20/wax patterns	CC-20/wax	7.81-40.00	9.02-40.00	0.19	11.00	-34.50	No	No	S28
PEEK/CIPs metastructure	PEEK/CIPs	5.10- 16.80,18.25- 40.00	7.70-14.70, 22.60- 36.30	2.90	10.00	-30.50	8.46	No	S29
CNFs-SiBCN-based metamaterial	CNFs-SiBCN	7.60-40.00	17.00-37.00	0.40	5.00	-46.00	No	No	S30
Nested triple-helix	Fe single-helix	11.50-13.75 15.00-15.50 18.70-40.00	11.67-13.30 15.00-15.05 20.50-40.00	0.00	15.00	-42.50	No	No	S31
Graphene foam	Graphene Foam	4.10-18.00; 26.50-29.80;	5.40-15.00	0.90	10.00	-35.00	No	No	S32
Fe/SiC-C fiber superstructure	Fe/SiC-C fiber	2.80-18.00	4.80-11.20	5.20	8.45	-24.00	No	No	S33
CuHT-FCIP-EP metamaterial	CuHT-FCIP 2D/2D assembly	2.00-40.00	2.00-2.35 4.50-40.00	6.00	9.30	-51.00	201.01	0.89	This work

5. What is the thickness of samples S1, S2 and S3 in CuHT-FCIP-EP respectively? Because this thickness has a great influence on the absorption bandwidth in 3D CuHT-FCIP-EP. So what is the effect of the introduction of different thicknesses on the absorption bandwidth of 2D CuHT-FCIP-EP? Can the authors add a case of thickness effect in addition to examples of different CuHT/FCIP ratios?

Response: Thanks for the comment. The thicknesses of the sample S1, S2, and S3 used in the

metamaterial were 2.0 mm (top layer), 3.6 mm (middle layer), and 3.7 mm (bottom layer), respectively. The effect of different thicknesses on the absorption bandwidth of these three 2D composites can be found in **Figure 3a-c**. Just like typical dielectric materials, as the sample thickness increases, the position of maximum absorption of S1-S3 samples under the metal backplane model gradually shifts towards the lower frequency region. This is because the reflective loss of dielectric-type absorbers under the metal backplane model follows the 1/4 wavelength theory, meaning when the thickness of the sample is 1/4 of the wavelength of the electromagnetic wave inside the sample, the sample exhibits the strongest loss of electromagnetic wave. The optimal EAB of S1-S3 is 5.44 GHz, 6.16 GHz, and 5.30 GHz at thicknesses of 1.90 mm, 1.78 mm, and 1.43 mm, respectively. This is mainly because, first, compared to S1, both S2 and S3 contain a larger amount of magnetic components (FCIP), resulting in a higher overall refractive index ($n = \sqrt{\epsilon\mu}$); second, sample S2 exhibits a magnetoelectric coupling loss exactly in the 6-18 GHz range, thus enhancing its absorption capability within this range. Meanwhile, at the same thickness, the strongest absorption peak frequencies of S1-S3 decrease sequentially. For example, when the thickness is 4 mm, the minimum absorption peak position for S1 is 5.82 GHz, for S2 it is 5.03 GHz, and for S3 it is 3.96 GHz.

In order to better visually display the reflection loss of these three materials at different thicknesses under the metal backplane model, we transformed the data in **Figure 3a-c** into separate curves, shown in **Figure S9**. Corresponding descriptions have been added as well.

Figure S9. Reflection loss of 2D CuHT-FCIP-EP at different thicknesses for (a) CuHT-FCIP-EP-S1, (b) CuHT-FCIP-EP-S2 and (c) CuHT-FCIP-EP-S3.

SI (Figure S9)

“The CuHT/FCIP ratio significantly impacts both the position of the minimum absorption peak and the effective absorption bandwidth (EAB) of the composite materials. For dielectric-type electromagnetic

absorption materials, the thickness of the absorber and its strongest absorption frequency satisfy the following equation:

$$d = \frac{c}{4f\sqrt{\epsilon_r\mu_r}} \quad S3$$

where d is the thickness of the absorber, ϵ_r is the complex permittivity, μ_r is the complex permeability, f is the frequency, and c is the speed of electromagnetic waves in vacuum. At lower thicknesses, sample S1-S3 exhibit significant absorption only in the high-frequency range (Ku band). As the thickness increases, the EAB gradually shifts towards lower frequency ranges. Meanwhile, at the same thickness, the strongest absorption peak frequencies of S1-S3 decrease sequentially. For example, when the thickness is 4 mm, the minimum absorption peak position for S1 is 5.82 GHz, for S2 it is 5.03 GHz, and for S3 it is 3.96 GHz. Moreover, the optimal EAB of S1-S3 is 5.44 GHz, 6.16 GHz, and 5.30 GHz at thicknesses of 1.90 mm, 1.78 mm, and 1.43 mm, respectively. This is mainly because, first, compared to S1, both S2 and S3 contain a larger amount of magnetic components (FCIP), resulting in a higher overall refractive index ($n = \sqrt{\epsilon\mu}$); second, sample S2 exhibits a strong magneto-dielectric coupling loss exactly in the 6-18 GHz range, thus enhancing its absorption capability within this range. Hence, the ratio of CuHT to FCIP significantly influences the absorption performance of the composite materials, providing a material foundation for the design of multilayer metamaterials.”

6. What does "t FCIP" mean on page 10, line 15?

Response: We sincerely apologize for this typo. The correct sentence is updated.

Page 10

“To further explore the effect of FCIP content on the EMW absorption performance of the CuHT–FCIP–EP composite system...”

7. Last sentence on page 15 "Furthermore, to the best of our knowledge, its -15 dB EAB (frequency ranges when $RL < -15$ dB), 35.85 dB, is also found to be one of the trade-off among the three-month (Figure S12) ". What does it mean to have a -15db absorption bandwidth of 35.85dB?

Response: Thank you for pointing this out. “35.85 dB” should be “35.85 GHz”, which means the frequency range when $RL < -15$ dB is 35.85 GHz. We apologize for the oversight in the previous manuscript. Below is the updated sentence:

Page 16

“Furthermore, to the best of our knowledge, its -15 dB EAB (frequency ranges when $RL < -15$ dB), 35.85 GHz, is also found to be one of the broadest among the literatures (Figure S15).”

We hope the point-by-point response to reviewer can address the concern of reviewers. If any questions, do not hesitate to contact us. And we will surely revise it again as you request. Once again, we sincerely appreciate your and reviewers' warm work and look forward to your reply.

Yours sincerely,

Jie Kong

Professor, Humboldt Follow, Clarivate Analytics Highly-Cited Researcher
School of Chemistry and Chemical Engineering, Northwestern Polytechnical University
E-mail Address: kongjie@nwpu.edu.cn

REVIEWERS' COMMENTS

Reviewer #1 (Remarks to the Author):

This manuscript has been carefully revised according to referee's comments. I would like to recommend the acceptance of it for publication in NC.

Reviewer #3 (Remarks to the Author):

The authors gave a positive reply to the reviewers' comments. Some new analysed have been added in the revised manuscript. It is recommended for publication in present form.